# Edible Coatings and Future Trends in Active Food Packaging–Fruits’ and Traditional Sausages’ Shelf Life Increasing

**DOI:** 10.3390/foods12173308

**Published:** 2023-09-02

**Authors:** Catarina Nunes, Mafalda Silva, Diana Farinha, Hélia Sales, Rita Pontes, João Nunes

**Affiliations:** Association BLC3–Technology and Innovation Campus, Centre Bio R&D Unit, Rua Nossa Senhora da Conceição 2, Lagares da Beira, 3405-155 Oliveira do Hospital, Portugal; catarina.nunes@blc3.pt (C.N.); mafalda.silva@blc3.pt (M.S.); helia.sales@blc3.pt (H.S.); rita.pontes@blc3.pt (R.P.); joao.nunes@blc3.pt (J.N.)

**Keywords:** antimicrobial compounds, bionanocomposites, circular bioeconomy, consumable films, food waste, food perishability delay, natural antimicrobial agents

## Abstract

The global food production industry faces environmental concerns exacerbated by substantial food waste. European countries are striving to reduce food waste towards a circular bioeconomy and sustainable development. To address environmental issues and reduce plastic waste, researchers are developing sustainable active packaging systems, including edible packaging made from industry residues. These innovations aim to increase food safety and quality, extend shelf life, and reduce plastic and food waste. Particularly important in the context of the growing demand for fresh and minimally processed fruits, edible coatings have emerged as a potential solution that offers numerous advantages in maintaining fruit quality. In addition to fruit, edible coatings have also been investigated for animal-based foods to meet the demand for high-quality, chemical-free food and extended shelf life. These products globally consumed can be susceptible to the growth of harmful microorganisms and spoilage. One of the main advantages of using edible coatings is their ability to preserve meat quality and freshness by reducing undesirable physicochemical changes, such as color, texture, and moisture loss. Furthermore, edible coatings also contribute to the development of a circular bioeconomy, promoting sustainability in the food industry. This paper reviews the antimicrobial edible coatings investigated in recent years in minimally processed fruits and traditional sausages. It also approaches bionanocomposites as a recently emerged technology with potential application in food quality and safety.

## 1. Introduction

As the world’s population continuously increases, so does consumer demand for fresh, healthy and nutritious food products, aligned with food short shelf-life, resulting in a huge amount of food waste. In fact, this is a global problem and thus turned food waste reduction into a fundamental challenge [1].

Food stability and consequent shelf-life relies on a combination of different factors, namely the quality of the ingredients, structure, composition, processing, manufacturing conditions, and associated packaging system [2].

The shelf life of a food product is understood as the time period in which a food product is safe while maintaining its desired sensory, chemical, physical, microbiological, and functional characteristics unaltered, in accordance with labeling information when stored in the recommended conditions. 

Increasing food products’ shelf-life results in reduced food and package waste, as food maintains its original characteristics for longer.

Fruits are known for their excellent nutritional intake, namely the abundance in fibers, minerals, and vitamins. They are mainly composed of living tissues and characterized by a high level of water availability, which make them susceptible for microbial contamination and, therefore, rapid deterioration. Increasing shelf-life and maintaining the intrinsic properties represents a major challenge for a suitable fruit packaging system [1]. On the other hand, meat products represent the main source of protein around the world, including cured meat products, such as traditional sausages. The packaging sector faces a challenge in this range of products, as it is prone to microbial growth, especially for pathogens such as *Listeria monocytogenes.* Recontamination also represents a major issue in various ready-to-eat meat products [1]. This review focuses on these two groups, fruits and traditional sausages, due to their importance and sensibility.

Recent advances have been made in order to create new sustainable solutions in food packaging systems. Edible packaging systems have the potential to meet sustainability needs and can contribute to the reduction of plastic waste, antimicrobial agents, and deterioration. However, it must be considered a safe strategy to reduce the exposure of consumers to health dangers associated with the use of such substances. Bionanocomposites produced from biocompatible and antimicrobial substances can help to reach a balanced solution to meet the presented needs: shelf-life extension, health safety, food, and plastic waste reduction [3]. It arises as a novel technology offering controlled release and stability of biomolecules, increasing solubility, stability, and bioavailability of bioactive compounds. Additionally, it contributes to the preservation of food’s inner properties and helps prevent loss of functionality of the coating systems [2].

The antimicrobial edible coating is approached in this paper, starting from the ecological and associated bio-economical potential point of view and focusing on its main characteristics and applications, in particular of minimally processed fruits and traditional sausages. Finally, it addresses bionanocomposites as an emergent technology and its potential in food coating applications.

### Food Waste and Circular Bioeconomy

As the global population increases, so does food production. However, this growth poses a threat to climate stability and ecosystems while also boosting environmental degradation [4]. According to FAO, there are 811 million people with no access to food and 132 million suffering with food insecurity, and still, 931 million tons of food are wasted yearly, along with all the resources, such as water, soil, energy, and human work used in the production and distribution stages [5]. In 2050, the population is expected to reach 9.7 billion, posing an even bigger threat to a planet that is already exceeding its limits and where the effects of climate change are undeniable [4].

Packaging is one of the principal products value chains considered in the action plan for circular economy for a cleaner and competitive Europe. In 2017, packaging residues reached 173 kg per capita, the highest value so far. The strategy lays on the packaging excess and waste reduction and the reutilization and recycling of materials [6]. As aesthetic criteria and food product durability are among the main driving forces behind food waste [7], innovation in food packaging has the potential to contribute for food waste reduction and thus help to reach the European Commission Sustainable Development Goals.

Aligned with the European Union’s (EU) objective of attaining climate neutrality by 2050 through the Green Deal initiative, the European Commission (EC) introduced the initial set of measures in March 2020 to expedite the shift towards a circular economy, as laid out in the Circular Economy Action Plan. In this context, the Commission introduced fresh rules across the EU concerning packaging. These proposals involve ideas to improve packaging design—like clear labeling—with the aim of encouraging the reuse and recycling of packaging. Moreover, they require a transition towards plastics made from biological sources that are capable of breaking down naturally, either through biodegradation or composting.

Microbiological contamination is responsible for the loss of more than 25% of the food before consumption [8,9]. Packaging plays an important role in this regard, preventing food deterioration during storage, transportation, and distribution processes. The basic purpose of packaging-protection, containment, information, and convenience evolved from the simple barrier function to new functional active packaging systems as modified atmosphere packaging and edible films or coatings [10]. Active packaging implies the interaction between the package, the product, and the environment [11,12,13,14,15]. Furthermore, the food industry is currently dealing with an increasingly more informed and demanding market, which boosts the creation of new solutions to respond to consumers demands for quality, safety, and sustainability. Non-biodegradable packaging materials have noteworthy disadvantages representing a threat to the ecosystems due to higher levels of toxic emissions, modifications in the carbon dioxide cycle, and composting problems [16]. Thus, researchers’ focus has shifted to sustainable advances in active packaging systems to preserve food quality and sensory characteristic over time [3]. Additionally, according to Portuguese Environmental Agency, the economy is currently shifting towards a circular sustainable bioeconomy model, with a focus on the sustainable production and the intelligent use of regional biological resources and promoting research, promoting innovation, and raising awareness of the importance of such a transition and to generate knowledge on this subject [17]. The incorporation of active chemicals extracted from industrial wastes into edible films is a trending topic in material research with an increasing number of results [18]. Several steps have been taken towards the use of industry residues as feedstock to produce food edible packages with the potential to feed a new economical field [19,20,21,22,23,24,25,26,27,28,29]. The result should be a reduction of the general environmental impact of food industrial activity, not only for turning residues into subproducts for the potential shelf-life extension, but also for the reduction of plastic waste. The referred studies focus on exploring different materials and techniques for the development of antimicrobial and biodegradable edible films and coatings for food packaging. These studies highlighted the potential of using cheese whey, lime essential oil, fermented whey, starch, bacterial cellulose, nanofibers, banana peel pectin, Mediterranean herbs, essential oils, chitosan, protein concentrate from shrimp waste, plant gum exudates, and bacterial cellulose-lactoferrin in the production of functional and sustainable packaging solutions. The findings indicated that these materials and additives can enhance the mechanical properties, antimicrobial activity, and edible films shelf-life extension. Moreover, the use of essential oils and bioactive compounds from natural sources offered the advantage of improving the sensory characteristics and preserving the quality of food products. Overall, the research suggested that edible films and coatings derived from these materials have the potential to provide effective and environmentally friendly alternatives to traditional packaging methods, contributing to the reduction of plastic waste and promoting food safety and quality.

The bionanocomposites arise as a new technology that presents several advantages in terms of biomolecules application in coating systems; it also contributes to the preservation of food properties, such as flavor and aroma and also improving barrier functions.

It should be noted that both edible coatings and new active food packaging have an impact on increasing the shelf life of food and enable a reduction in food waste generation, thus contributing to the Circular Bioeconomy.

## 2. Edible Coatings (General Concept)

Currently, consumers are looking for safer and more natural foods, namely foods free of chemical and synthetic preservatives. Accordingly, to address the need for safer and healthier food options, the creation of edible coatings enriched with bioactive compounds has emerged. These coatings are made from biodegradable and bioactive ingredients, that can delay food ripening, meeting this demand for safer and healthier food and ensuring that consumers can enjoy the food without any health danger associated.

Edible coatings are carefully applied onto the food products’ surface and reduce food degradation as they form a barrier that prevents or regulates the transference of gases and moisture from the atmosphere, reducing food cellular respiration and microbial propagation. In addition to delaying ripening, edible coatings reduce the use of synthetic conventional packaging, reducing waste related to the food industry [30].

Over time, numerous edible coatings containing diverse compounds have been suggested. The components used to create these coatings and films, which are both edible and biodegradable, can arise from a wide range of natural origins and are characterized by their structural complexity and functional diversity. They are classified as: polysaccharides, proteins, and lipids. 

Coatings can also incorporate food additives, including organic acids, enzymes, bacteriocins, fungicides, natural extracts, vitamins, etc. that can extend product shelf-life by reducing the risk of pathogen growth on the food surface and, additionally, improve the sensory quality of the packaged or coated product [31]. Once the edible coating is formulated to suit the specific requirements of the food product, it becomes essential to study the effects of adding certain compounds to it. This is crucial because the introduction of additional ingredients could potentially modify the fundamental functional characteristics of the coating. The impact of an ingredient on the coating’s functionality is influenced by factors such as its concentration, stability, chemical structure, how well it disperses within the coating, and how it interacts with the polymer used in the formulation [32]. In the literature, there are various edible coatings that help increase the shelf life of food, as shown in Figure 1.

Polysaccharides are stable and non-toxic substances used to produce edible coatings/films with great potential to formulate films around food products and include starch, alginates, carrageenan, chitosan, and gums. Natural gums can be obtained from seaweed extracts (alginates, agar), seed gums (galactomannans), or roots, for example. Coatings with polysaccharides ease the addition of bioactive compounds to induce a superior delay in food deterioration, such as ascorbic acid, citric acid, lemongrass oil, cinnamon leaf or bark oil, and coconut oil. Starch is abundant in foods, such as cereals and vegetables, making it a low-cost and biodegradable polysaccharide, and it prevent food contact with the atmosphere. Starch has been applied as a coating for strawberries [35,36], apples, pineapples [36,37], and mangoes [38], showing posistive results in food storage. 

Cellulose is also a promising polysaccharide for application in food coatings despite its insolubility in water, which aport some limitations to the use of this polysacaride in this context. Cellulose derivatives, such as carboxymethylcellulose (CMC), methylcellulose (MC), hydeoxyprpoylcelluose (HPC), and hydroxypropylmethylcellulose (HPMC), can be a viable application of cellulose materials [39]. Moreover, cellulose gums have been effectively used for over twenty years to prevent the ripening of fruits, such as bananas, papayas, and mangoes [40].

Alginate is a compound isolated from algae, with the ability to create translucent coatings with a shiny appearance. Alginate coatings have been tested on ‘Gala’ apples, papayas, melons, and pears [41], and it has been observed that ripening is delayed, preventing fruit dehydration.

Proteins have the ability to create edible coatings, which can be derived from animal and plant sources. Animal-based proteins includes those found in whey, casein, collagen, or gelatin. Plant-based proteins include zein from corn, gluten from wheat, soy protein, pea protein, rice bran portein, rice seed protein, and peanut protein. Protein-based coatings are good barriers to gases but less effective in food waterprooofing [30]. Casein, a whey protein, is a good coating material for food as it has high nutitional quality, thus adding a nutritional value to its coating purpose. Zein from corn has been widely used to coat fruits and vegetables as well as nuts, apples, and pears, delaying weight loss during storage. 

When used as a coating for food, lipids prevent water loss in the food, as these molecules have a low affinity for water, resulting in an improved quality and appearance. Edible lipids commonly used include beeswax, candelilla wax, carnauba wax, and fatty acids [30]. Carnauba wax, a vegetable wax, does not provide shine or effectively isolate the food from the atmosphere, but it acts as an excellent barrier against moisture. It is often used in combination with other coatings, such as shellac, to coat apples and citrus fruits. Beeswax has been used in combination with other coatings, such as pea starch and cellulose derivatives, to create more compact and thicker bio coatings [42]. Beeswax combined with protein-based formulations has been applied to preserve plums.

There are different methodologies used for coating creation depending on the long-term conditions in which the food will be exposed, these conditions must be the same for both the food product and the coating. The coatings can be applied in food through drying, heating, cooling, or coagulation [43]. The application method can influence the effectiveness of the coating, so it must be chosen appropriately to achieve the most promising results. According to the state of the coating material, which can be a liquid, a suspension, an emulsion, or a powder, there are several techniques for its application, further described.

Immersion

Immersion is the most commonly used method as it is easy to apply and more cost-effective. In this methodology, the coating is in a liquid, suspension, or emulsion state and involves three distinct steps: immersion and dwell time, deposition, and solvent evaporation [44]. The first step is to immerse the food in the coating at a constant velocity so that the food comes into contact with and traps the coating on its surface. Deposition is a step performed to obtain thin layers of coating, where after immersion, the food is kept stationary to allow excess coating to be removed by deposition [45]. The final step, solvent evaporation, involves the evaporation of coating excess, which can be achieved through drying equipment or simply by ambient temperature to facilitate evaporation and drying of the food. The immersion time can diverge from 5 to 30 s depending on the food product and edible coating purpose [46].

This method has many advantages, as the entire surface of the food is uniformly coated, and it can be applied to different surfaces. However, it has one disadvantage; if the coating becomes too thick, it can hinder the respiration of the food and potentially accumulate unwanted residues, leading to the development of microorganisms within the coating itself [47].

Spraying

Spraying is a method in which coating droplets are dispersed onto the food surface. There are several types of spraying, such as air spray atomization, which involves low-speed and cylindrical spray flow, air-assisted airless atomization, used for highly viscous coatings, and pressure atomization spraying, where coatings are applied using pressure instead of air [47].

In spraying, the pressure, viscosity, surface temperature, and coating tension are crucial aspects that significantly impact the final coating performance [47,48]. One advantage of spraying is the ability to apply two or more different coatings, creating a multi-layered coating [47]. Additionally, the thickness of the coating layer can be controlled through this technique.

Pan coating method

The pan coating methodology involves placing the food items inside a rotating container, where the coating is then sprayed or sprinkled, ensuring a homogeneous incorporation onto the food surface. Subsequently, the coating is dried [49].

The pan coating method is used to create thin or thick layers of coatings on hard surfaces, allowing for the simultaneous coating of a large number of food items, even with size variability and produces a transparent, flexible, and shining coating material. The main disadvantage of this technique lays on the fact that it takes some time to accomplish, considering the need for water evaporation during the process. Moreover, the coating must be added gradually; otherwise, it may cause the food items to stick together inside the rotating container [50].

### 2.1. Antimicrobial Edible Coating

The incorporation of antimicrobial substances into packaging materials has significant potential to enhance food safety and quality, leading to a shelf life extension for food products. This antimicrobial function can be achieved by including antimicrobial agents in the packaging or using antimicrobial polymers that meet standard packaging requirements, effectively restricting or preventing the growth of specific or broad range of microorganisms [51]. In the formulation of edible coatings, various antimicrobials are being considered to inhibit the growth of spoilage-causing microorganisms and reduce the risk of pathogens growth. Both natural and synthetic compounds are used to create antimicrobial coatings for food; however, there is a growing tendency for the select of natural antimicrobials sources and also for compounds that are generally recognized as safe (GRAS), aligning with consumer demands for healthy food without chemical additives [52]. Commonly, antimicrobials used in edible films and coatings includes organic acids, chitosan (a polysaccharide), polypeptides such as nisin, the lactoperoxidase system, and plant extracts and essential oils, among others [53]. Researchers are exploring natural antimicrobial substances to incorporate into edible films and coatings safe for consumption, meeting the consumer demands for chemical-free and healthy food options. When choosing an antimicrobial agent for edible films and coatings, it is important to consider its effectiveness against the specific microorganism targeted as well as any potential interactions between the film-forming biopolymer and other components present in the food. These interactions can influence both the antimicrobial activity and the film properties, making them crucial factors in the development of effective antimicrobial films and coatings [53]. The use of inherently antimicrobial polymers is also becoming increasingly important [32]. Bacteriocins offer an intriguing option as antimicrobial compounds since they act as natural preservatives, eliminating the need for synthetic additives in food.

For centuries, spices and herbs have been recognized for their antimicrobial properties and have been commonly used as seasoning additives in food due to their aromatic properties [54]. Numerous studies have highlighted the preservative effects of plant-derived compounds in various food applications, along with the factors that influence their efficacy. Essential oils and their major constituents, such as thymol, carvacrol, p-cymene, and γ-terpinene found in *Thymus* species [55], have garnered attention due to the presence of phenolic compounds or other hydrophobic components [56]. However, the effectiveness of these compounds against pathogens can vary significantly due to their structural diversity and variations in chemical composition. For instance, the antimicrobial properties of the methanolic extract from *Mosla chinensis* were evaluated against eight bacterial and nine fungal strains [57]. The essential oil, which contains carvacrol (57%), p-cymene (14%), thymol acetate (13%), thymol (7%), and γ-terpinene (2%) as its main components, exhibited considerable potential against two Gram-positive bacteria commonly found in many food products, namely *S. aureus* and *L. monocytogenes* [58]. Essential oils, in general, contain a variety of compounds, such as monoterpenes, sesquiterpenes, esters, aldehydes, ketones, acids, flavonoids, and polyphenols, which are widely recognized for their antimicrobial properties [59]. The exact mechanism of action against bacteria is not fully understood because each compound within an essential oil has its own unique antimicrobial mechanism, which may be effective against specific types of microorganisms found in certain food products [60]. 

The use of plant extracts, natural volatile compounds, and essential oils in food coatings has been found to reduce food spoilage by harnessing their antimicrobial capabilities. Fresh foods, particularly, are exposed to various microorganisms present in the soil, air, and the plant itself, which can lead to food contamination and decay. Plant extracts with antimicrobial properties are considered non-toxic biopreservatives and offer an easy application method. Numerous studies in the literature have shown positive effects of plant extracts or combinations of extracts from different plants in extending the shelf life of post-harvest foods. For example, a combination of garlic and ginger extracts sprayed on tomatoes inhibited the growth of bacteria and fungi, except for *Rhizopus* and *Aspergillus*, thus delaying decay [61]. Turmeric extract combined with ginger was also found to have inhibitory effects on the growth of fungi, such as *Aspergillus niger* and *Penicillium digitatum* [62]. Natural volatile compounds, including methyl jasmonate, ethanol, tea tree oil, and garlic oil, when applied to tomatoes, have shown a reduction in microbial proliferation while preserving fruit characteristics, such as color and firmness. Other volatile compounds such as lemongrass oil, oregano oil, and vanilla are added to coatings to enhance their properties and inhibit the growth of fungi, molds, and yeasts.

Essential oils and phenolics, commonly used in diets like the Indian diet, are recognized for their potential health benefits. Compounds like vanillin possess antioxidant properties, which aid in preserving the qualities of coated foods. *Verbena officinalis* and *Origanum vulgare* oil inhibit fungal growth, making them potential substitutes for synthetic fungicides. Oils extracted from oregano, rosemary, thyme, and sage are effective in inhibiting gram-positive bacteria activity [63]. Essential oils are not directly applied to foods but are incorporated into other coatings, such as polysaccharide-based coatings. For instance, cinnamon and lemongrass essential oils were added to an alginate coating and applied to melons, resulting in a shelf-life extension of 21 days [64]. *Thymus vulgaris* essential oil reduced fruit degradation by exhibiting antifungal activity against *Botrytis cinerea*, *Phytophthora citrophthora*, and *Rhizopus stolonifera* [65]. Similarly, a pectin coating enriched with cinnamon leaf essential oil reduced bacterial proliferation in peaches [66].

Certain polymers derived from animals, including chitosan, whey proteins, protein hydrolysates, and bioactive peptides, possess natural antimicrobial properties. Chitosan, in particular, has shown an inhibitory effect on various bacteria, molds, and yeasts. Its mode of action involves interacting with the cell membranes of microorganisms, causing the formation of holes and leakage of cell contents. Chitosan’s film-forming ability further enhances its applicability in different contexts. Another interesting characteristic of chitosan is its metal-chelating capability near bacteria, which hinders the movement of essential nutrients and inhibits bacterial growth. Factors such as pH, targeted microorganisms, exposure time, degree of acetylation, and the positive charge of chitosan influence its antimicrobial activity [67]. On the other hand, whey protein, a by-product of the dairy industry, also exhibits high biological activity and contains bioactive peptides with antimicrobial effects. However, the exact mechanism of action for these peptides is still not fully understood. Similar to chitosan, the antimicrobial activity of whey protein is influenced by factors such as pH, temperature, and the presence of fat. In previous studies [68], whey protein nanofibrils combined with titanium dioxide nanotubes were incorporated into edible films to investigate their antioxidant and antimicrobial effects on chilled meat storage. The results indicated that the whey protein-containing films exhibited inhibition zones with diameters greater than or equal to 10 mm against bacteria, such as *Listeria monocytogenes, Staphylococcus aureus*, *Escherichia coli*, and *Salmonella enteritidis*. This suggests that the films containing whey protein contributed to an extended storage life for chilled meat [68]. In Figure 2 some examples of antimicrobial agents that can be incorporated into edible coatings for food applications are presented.

### 2.2. Characteristics of Antimicrobial Coatings Designed for Food Packaging Use

The antimicrobial edible films and coatings have proven to be efficient in safeguarding food against spoilage and inhibiting the proliferation of harmful microorganisms. This is achieved by regulating the controlled release of antimicrobial agents onto the food’s surface [70]. When creating these films and coatings, careful consideration is given to selecting suitable antimicrobial agents based on their effectiveness against specific target microorganisms and their compatibility with the film-forming polymers and the packaged food. These interactions are vital as they significantly influence both the antimicrobial activity and the film properties [53]. To evaluate the mechanical and barrier properties of edible films and coatings, various physical tests are performed. Tests such as quasi-static tension or puncture tests help assess mechanical parameters such as elastic modulus, tensile strength, and strain at break [71]. Water vapor permeability (WVP) is determined using the ASTM E-96 static method, which is crucial for applications involving aqueous food storage. In some cases, high water solubility is desirable, especially when the film or coating is meant to be consumed with the food. However, certain biopolymer-based edible films and coatings have limited applications in food packaging due to their poor water vapor barrier and low mechanical strength [72].

To address these limitations, researchers have proposed solutions such as blending with different biopolymers or incorporating hydrophobic materials, such as oils or waxes, to improve crosslinking and enhance their properties [73,74,75]. For instance, Gutiérrez et al. improved the hydrophilic characteristics and degradation temperature of starch-based edible films by cross-linking them using sodium trimetaphosphate, adhering to safety standards set by the U.S. Food and Drug Administration (FDA) [76].

Other studies, such as those conducted by Schmid et al., revealed that the extent of denaturation in whey protein isolate-based coatings significantly affected cross-linking density, which, in turn, impacted the number of disulfide bonds in the network [77].

### 2.3. Applications of Antimicrobial Edible Coatings: Fruit and Traditional Sausages

Over the last few years, there has been a notable rise in fruit consumption, resulting in a significant upsurge in worldwide fruit production. In 2022, the global consumption of fresh fruit amounted to around 246billion kilograms, which marked a rise of approximately 7billion kilograms compared to the previous year. This upward trajectory is projected to persist in the following years, with estimates indicating a consumption of over 303.5 billion kilograms by the year 2028 [78]. Nevertheless, fresh fruits face substantial losses during production and preservation. To maintain their quality during the postharvest preservation stage, various technical treatments are used. One popular technique is applying edible coatings, which can be used on a wide variety of fruits to control the exchange of moisture and gases between the fruit and its surroundings. These coatings also offer an important advantage, as they allow for the incorporation of different active ingredients into the coating’s structure. This means that these substances can interact with the fruit and may even be consumed along with it. As a result, the fruit’s sensory and nutritional attributes are enhanced, and its shelf life is prolonged [79,80].

Regarding traditional sausages, characterized by a complex composition consisting of water, proteins, and lipids, they are prone to sensory and nutritional deterioration during processing and storage [81]. One common issue in traditional sausages is nonmicrobial spoilage caused by oxidation, which can be mitigated by using chemical substances, such as sulphur dioxide or synthetic antioxidants (e.g., butylated hydroxytoluene—BHT) to prevent oxidative reactions in lipids and proteins [81,82]. Additionally, traditional sausages provide a favorable environment for the growth of pathogenic and spoilage microorganisms due to their nutrient-rich composition. However, to counteract the growth of harmful microbes, traditional sausages often undergo a curing process, where nitrite salts are added to hinder microbial proliferation [83,84]. Nonetheless, numerous studies have established a connection between elevated consumption of these artificial preservatives and potential carcinogenic and allergenic effects [82,83,85]. As a result, there is a growing demand for meat products free from synthetic chemical preservatives. This has led to an increased use of additives derived from natural sources, which have gained popularity as effective means of prolonging the shelf life of food items due to their antimicrobial and antioxidant properties [86,87,88]. According to this, in recent years, researchers have focused on studies applying edible coatings with active compounds in this type of food products, aimed at improving quality, extending shelf life, and reducing the use of synthetic preservatives.

#### 2.3.1. Minimally Processed Fruit

The consumption of minimally processed foods, such as peeled and pitted packaged fruits, has been on the rise over the last few years. The demand from consumers for this type of product, along with the increased time from “farm to plate,” requires the development of technological solutions that maximize shelf life without compromising their attributes [89].

Minimally processed fruits, also known as “fourth range” or fresh-cut fruits, are highly vulnerable and susceptible to rapid deterioration. There is often a considerable time gap between their harvest and consumption, which can compromise the quality of these foods. During this time gap, possible dehydration, deterioration, loss of aroma, decrease in nutritional value, and degradation of appearance can occur. These changes can take place in just a few hours if necessary measures are not taken after harvest [90].

In minimal processing, actions to eliminate microorganisms are limited. Therefore, measures that could be considered barriers or obstacles to the occurrence of microbiological contamination were developed. These measures included washing, the use of disinfectants, packaging in modified atmosphere, and refrigeration. However, the study of other barriers that could have contributed to maintaining the quality of the products was also necessary, such as freshness, aroma, color, and taste, without altering their nutritional and sensory characteristics. These food products were highly perishable due to the exposure of their internal tissues, leading to an acceleration in their metabolism as a consequence of the aforementioned physical alteration [91]. In other words, cutting and peeling caused injuries to the tissues of these products, which, combined with accelerated metabolism, greatly contributed to the loss of product quality, affecting its shelf life [92].

Packaging plays a crucial role in fruit preservation. However, the current trend in food packaging materials is dominated by synthetic and non-biodegradable petroleum-based polymers, which have a detrimental ecological impact. In contrast, edible coatings offer a more environmentally friendly solution as they are made from biodegradable polymers, allowing for the utilization of food industry by-products and reducing the need for conventional packaging materials [93]. 

The antimicrobial edible coatings made from polysaccharides, such as sodium alginate (NaAlg), chitosan (CH), and hydroxypropyl methylcellulose (HPMC), are commonly used to protect minimally processed fruits. These coatings have excellent film-forming properties and selectively allow for the passage of oxygen (O_2_) and carbon dioxide (CO_2_). They can help reduce the respiration rates of the product, delaying the ripening and senescence process similar to storage under modified or controlled atmospheres [94]. However, it is important to consider the potential accumulation of CO_2_ and depletion of O_2_ within the fruit’s internal atmosphere when using edible coatings. Excessive CO_2_ levels and low O_2_ levels can lead to anaerobic fermentation and the production of off-odors and off-tastes in the fruits. Therefore, the selection of a suitable coating material is crucial to maintain a desirable internal gas composition based on the respiration and transpiration rates of the specific fruit or vegetable. Additionally, controlling the wettability of the coating formulations is important as it can affect the thickness of the coating and its permeability [95].

Environmental conditions in the storage area, such as temperature and relative humidity, also play a significant role in the internal atmosphere of fresh fruits. These conditions can strongly influence the permeability of the coating and the respiration rates of the products. Therefore, it is necessary to carefully manage these environmental factors (permeability) [95].

Numerous studies have demonstrated the effectiveness of polysaccharide-based coatings in preserving the quality of fresh fruits. These coatings can act as carriers for natural antimicrobial substances, mainly derived from plants. They help reduce respiration rates, prevent weight loss, maintain texture and flavor, and inhibit the proliferation and metabolic activity of microorganisms. The incorporation of plant-derived antimicrobial compounds into the coatings has been shown to effectively reduce microbial contamination in post-harvest fruits and minimally processed products [96].

Edible coatings made from polysaccharides provide a protective barrier for fresh and minimally processed fruits and vegetables. They can control the internal gas composition, reduce respiration rates, and incorporate natural antimicrobial compounds, contributing to the preservation of quality and reduction of microbial contamination [96]. 

Edible coatings have emerged as a response to consumer demands for improved product quality and longer shelf life while simultaneously addressing the environmental concerns associated with traditional food packaging. By opting for edible coatings, the food industry can take significant steps towards reducing its ecological footprint. Edible coatings for minimally processed fruits provide an alternative to modified atmosphere packaging, reducing qualitative changes and consequent losses by controlling and modifying the internal atmosphere of each individual product [94]. Additionally, they offer a semipermeable barrier aimed at extending the shelf life of products. This is achieved by reducing moisture, solute migration, gas exchange, and respiration rate (due to selective permeability to oxygen and carbon dioxide) and oxidative reactions [94]. 

In Table 1, some example antimicrobial used coatings in fruit applications are presented.

#### 2.3.2. Traditional Sausages

Traditional sausages are considered stable products, taking into account the pH values and water activity that they normally present. However, this type of product also requires special attention with regard to the growth of spoilage microflora. During the storage period, there is a rapid colonization on the surface by a high number of filamentous fungi, molds and yeasts as well as the potential development of some pathogenic microorganisms [93].

The main factor that deteriorates the quality of this type of product is the presence of oxygen, which triggers oxidative rancidity and changes in taste, color, and the development of aerobic bacteria and fungi [116,117].

The development of filamentous fungi on the surface of traditional sausages is undesirable and can lead to an increase in the appearance of toxic secondary metabolites, called mycotoxins [93].

Failure to control fungal growth causes them to develop and produce a grayish or yellowish pigmentation in the food product. In addition, the toxins produced can, albeit rarely, spread through the meat. Fungal development in the mentioned product can occur during the distribution phase or just at the consumer’s home after opening the package. In this sense, it is essential to inhibit the development of said microbiota during its useful life. Accordingly, the need arose to study innovative techniques that lead to the inhibition of the development of fungi and consequently allow for extending the shelf life of traditional sausages [31,118,119].

After opening the package of traditional sausages, Portuguese consumers have the habit of storing them in the fridge wrapped in cling film [93].

With regard to improving the quality and increasing the shelf life of traditional sausages, these have received special attention from science and technology, especially in Central and South Europe and North America.

Various packaging methods and edible coatings with antimicrobial properties have been extensively researched. This innovative form of active packaging has been applied in the food industry since its inception, aiming to reduce, inhibit, or eliminate microbial growth on the food’s surface [93]. Minimally processed ready-to-eat meats, such as traditional sausages, can potentially harbor harmful bacteria like *Salmonella*, *Listeria*, and *E. coli*, which may contaminate the meat during processing or packaging. Studies indicate that a significant portion of foodborne illnesses is attributed to the consumption of contaminated meat products. To address this concern, scientists are investigating the use of edible films and coatings infused with antimicrobial agents. These coatings offer a promising solution in reducing the risk of harmful bacteria and extending the shelf life of meat products. Moreover, when applied to meat products, these coatings offer additional advantages to traditional sausages. In Table 2, some examples of the use of antimicrobial coatings are presented.

### 2.4. Edible Films and Coatings Obtained from Organic Food Residues–Examples of Waste Valorization and Circular Economy Potential in Food Packaging

Obtaining edible antimicrobial films and coatings from organic sources may be a sustainable path to prevent plastic waste; however, it must not burden agricultural practices. Otherwise, it could have impact in food products’ availability. Once we face an enormous amount of food waste scenario and also factory by-products, the focus must be driven to organic matter that already exists, thus lessening the waste problem instead.

Some authors already reported some food residue potential since fruit and vegetable waste is an appreciated source of natural products and bioactive compounds [52]. Plant sections usually seen as waste can be faced as raw materials in edible films production, such as skin, peel, seed, pomace, husk, and straw, among others, are bursting with polysaccharides and proteins, the main compounds used in edible film production. Apart from that, phenolic compounds can also be found and present properties such as antioxidant and anti-inflammatory activity, improving films functionality [135]. Tomato waste was reported as a rich source of phenolic compounds and antioxidant activity as well as flavoring compounds, thus an interesting source to be included in edible films and coatings. [136,137]. It was also reported that pumpkin waste can be a source of protein and pectin applied in film-forming materials [138]. Cellulose and lignin attained from forest residues may also be useful to strengthen biodegradable films [139,140]. An extract from a winery solid by-product was applied in poly(vinil alcohol)/gelatin films, and a flexibility improvement and antioxidant activity was observed in the films [141]. The potential use of mung bean protein and pomegranate peel to create edible films for food packaging was also explored [142]. Potato peel has the potential to form edible films, and the addition of curcumin presented a significant lipid oxidation reduction in fresh pork during storage [143]. Chitosan films supplemented with banana peel extracts applied as coating in apples showed antioxidant activity [109]. Fibers from mushroom by-products are related to enhancement of edible films and film properties [144].

Fermented whey proteins have been proposed as raw material to produce edible films, giving its main advantageous properties. It has bioactive peptides with antimicrobial activity, which also act as immunomodulatory agents that regulate cell-mediated and humoral immune functions, among another positive contributions to consumer health [145]. Besides its nutritional value, the use of fermented cheese whey proteins in coating or film production provides a good barrier and mechanical, functional, and antimicrobial properties [19], which make it a promising tool to achieve food packaging innovated systems, to reduce industrial waste, and to feed a circular economy and sustainability in the food industry.

## 3. Recent Developments in Food Packaging with Antimicrobial Properties

As the concern for sustainability and ecological safety grows, researchers have shifted their focus towards the development of easily degradable and biocompatible food packaging materials. Biopolymer-based packaging materials offer an advantage, as they can be disposed of in bio-waste decomposition centers, where they break down and release organic byproducts such as carbon dioxide (CO_2_) and water (H_2_O). However, despite their benefits, the use of biodegradable polymers in food packaging systems is sometimes limited due to various constraints.

Biodegradable polymers used in food packaging face certain limitations, such as having moderate mechanical barrier properties in comparison to synthetic polymers and less favorable thermal characteristics than conventional petroleum-based plastics. They may exhibit brittleness, low resistance to extended manufacturing processes, and limited melting resistance, melting enthalpy, and flexibility, which can make them less than ideal for food packaging applications [3]. However, despite these challenges, there have been significant efforts to enhance biopolymers to make them more suitable for use in food packaging systems. These endeavors are driven by the recognition of their ecological safety and sustainability advantages. Researchers are working to improve their properties and expand their potential in contributing to more environmentally friendly packaging solutions.

A recent and promising advancement in food packaging involves the application of nanotechnological concepts [146,147]. Nanotechnology-enhanced food packaging systems offer ecological advantages over conventional plastic barriers, and their functional components, such as antimicrobial agents, allow for extending food shelf life while detecting spoilage indicators such as off-flavors, color changes, and harmful food toxins. Smart and intelligent food packaging systems based on nanotechnology provide improved efficiency and enhanced food security [148]. Nanoparticles play a significant role as antimicrobial agents and carriers of diverse bioactive compounds. They actively release antioxidants, enzymes, flavors, anti-browning agents, antimicrobials, and other bioactive materials to preserve the food quality. This helps prevent microbial contamination, spoilage, and extends the product’s life even after the package is opened [149]. Certain metal and metal oxide nanoparticles, such as iron, silver, zinc oxides, carbon, magnesium oxides, titanium oxides, and silicon dioxide nanoparticles, are widely used as antimicrobials and food ingredients under specific conditions [150].

### Bionanocomposites and Issues Regarding Safety of Heavy and Nanoparticles

Bionanocomposites can be understood as multiphase materials composed of a biopolymer matrix and nanofillers (<100 nm) that offer structural integrity, barrier properties, and antimicrobial effects [151]. Nanofillers such as silicate, clay nanoplatelets, titanium dioxide, carbon nanotubes, chitin, graphene, cellulose-based nanofibers, and starch nanocrystals are incorporated into the biopolymer matrix to enhance thermal and gas barrier characteristics, provide fire resistance, and add functionality, such as antimicrobial and antioxidant effects, oxygen or moisture scavenging, and acting as biosensors to improve product shelf life [148,152,153,154,155]. Incorporating nanosized metals into biopolymer matrices provides stabilization and controlled antimicrobial action. For example, chitosan nanocomposite films with silver nanoparticles exhibit good antimicrobial activity [148]. Similarly, the addition of nano ZnO to PLA (polylactic acid) films enhances anti-UV and antibacterial properties [156]. Other metals and their oxide nanoparticles, including silver, copper, iron, gold, zinc, titanium dioxide, and palladium, have been studied for the development of antimicrobial and active packaging systems [157]. These nanoparticles generate reactive oxygen species, create electron–hole pairs under light irradiation, and bind metal ions to microbial membranes, leading to bacterial inactivation [158]. The antibacterial mechanisms of nanoparticles involve disrupting transportation; impairing DNA, RNA, and protein synthesis; and causing cell lysis. Gold nanoparticles can disrupt metabolism and tRNA assembly [157]. Overall, bionanocomposites offer promising opportunities for the development of advanced food packaging materials with improved properties and functionality. However, careful consideration of nanoparticle dispersion and potential toxicity is necessary to ensure their safe and effective use in food packaging applications [3].

Nanofillers (<100 nm) are widely used to enhance biopolymer matrices in food packaging, improving thermal barrier, and mechanical properties [152,159]. Careful design is crucial to withstand thermal and mechanical stresses [154,160]. Low concentrations of nanofillers (<5%) show significant improvements while higher concentrations enhance strength but reduce flexibility [151,154,161]. Various nanofillers such as silver nanoparticles [162], cellulose nanofibrils [163], zinc oxide nanorods [164], carbon nanotubes [151], and cellulose-based nanofibers starch nanocrystals are incorporated for enhanced food packaging [153,155]. Commonly investigated nanofillers include metallic, metal oxides, natural biopolymers, inorganic/organic materials, or natural antimicrobial substances. Bionanocomposites offer eco-friendly alternatives to conventional plastics and biopolymers in food packaging [3].

Nanoencapsulation systems have revolutionized food packaging by incorporating natural antimicrobial substances to protect food products from biological changes and environmental stresses during storage and distribution [148,157]. This process involves creating nano-sized particles through a two-step process, enhancing the system’s functionality by improving adsorption, solubility, bioavailability, and controlled release of bioactive compounds [165,166]. Various nano-encapsulating structures such as nanoemulsions, biopolymeric nanocarriers, solid lipid nanoparticles, electrospin nanofibers, and nanoliposomes are used to develop active antimicrobial packaging systems with controlled release. These systems provide a targeted and sustained release of bioactive compounds, improving food quality and shelf life while ensuring precise release, holding significant potential for enhancing food safety and preservation [3]. 

Despite the numerous advantages, it is essential to take into account potential toxicological risks associated with the migration of heavy particles and nanoparticles. European Food Safety Authority [167] guidelines restrict silver migration in food packaging to 0.05 mg/L in water and 0.05 mg/kg in food [168]. Prioritizing consumer health and safety thorough investigations into the potential toxic effects of nanoparticles in food packaging materials are necessary. By addressing these concerns, appropriate strategies can be developed to minimize migration and guarantee the overall safety and well-being of consumers [3]. Substances that either do not migrate or migrate in insignificant amounts have been approved for further use [167]. However, studies conducted by Lee et al. revealed that *in vivo* exposure to silver nanoparticles could lead to immunotoxicity and neurotoxicity by affecting genes related to immune cells, neurodegenerative conditions, and motor neuron disorders [169]. While silver nanoparticles have been used for an extended period, concerns persist about their potential entry and accumulation in vital organs, leading to increased production of reactive oxygen species [170]. Research has shown that these nanoparticles may cause cytotoxicity and genotoxicity and induce processes such as apoptosis, necrosis, and DNA strand breakage [171,172,173]. Consequently, a cautious risk assessment of nanoparticle materials poses a unique challenge to food safety [174,175].

## 4. Conclusions and Future Perspectives

Global food production and food waste, along with plastic waste, represent a challenge for environmental stability. Reducing plastic and food waste are part of the European commission agenda towards a Circular Bioeconomy and Sustainable Development.

Active packaging systems have the potential to contribute to a shelf-life extension, which will reduce food waste. On the other hand, biodegradable and antimicrobial edible films and coatings also support the plastic waste reduction initiatives. The increasing consumer demand for healthier food options has led to the use of antimicrobials derived from natural sources that can include waste valorization initiatives and represent a circular economy potential. It may represent a “win win” situation in terms of environmental stability and safety concerns. Minimally processed fruits and traditional sausages are among the most susceptible food products to be coated with antimicrobial edible films. Nanotechnology came along with new tools to improve food coating properties and has the potential to add on a positive nutritional effect on the package benefits with the inclusion of bioactive compounds, and nanoencapsulation systems have the potential to incorporate natural antimicrobial substances to enhance food shelf life. Despite its potential, toxicological risks associated with the migration of heavy particles and nanoparticles should be taken into account.

In 2017, the amount of packaging waste per person, known as packaging residues, reached a record high of 173 kg. Therefore, edible coatings and novel advancements in active food packaging can present a viable alternative to lower the historical values that have been reached.

## Figures and Tables

**Figure 1 foods-12-03308-f001:**
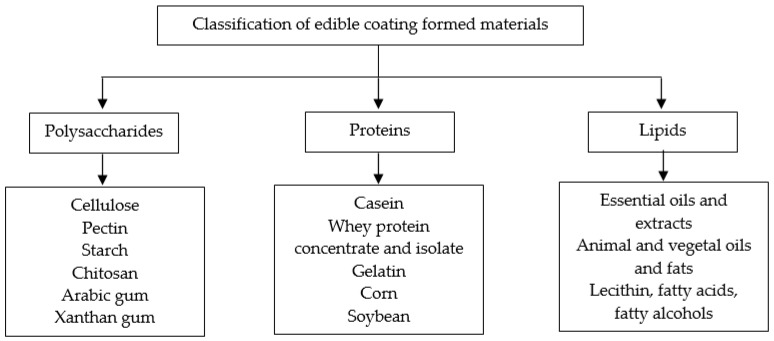
Classification of edible coating materials to apply in food products [33,34].

**Figure 2 foods-12-03308-f002:**
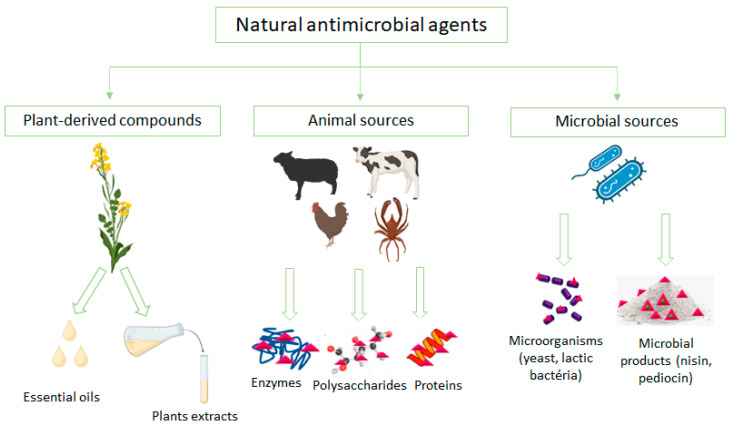
Natural sources for incorporating additives into edible packaging systems for food [69].

**Table 1 foods-12-03308-t001:** Edible coating with antimicrobial active compounds to apply in fruits and its advantages.

Fruit	Material Coating	Active Compounds	Coating Advantages	References
Apple	Soy protein Alginate Xanthan gum Carboxymethyl cellulose Chitosan	Thyme essential oil Cinnamon bark Tocopherol Ferulic acid *Aloe vera* Ascorbic acid Grape seed essential oil	Prevented water loss, showed antioxidants and antimicrobial properties, possess effective barrier properties and exhibit favorable mechanical and structural characteristics, physicochemical properties reserved, activity of PAP (phosphatidic acid phosphatase) and PPO (polyphenol oxidases) enzymes reduced and reduction in respiration rate observed, preserved the quality of food products, prolonging their shelf life and inhibiting browning.	[97,98,99,100,101,102]
Blackberries	Starch	Nystose Acetic acid or lactic acid	Antifungal effect, had beneficial impacts in slowing down pH increase, preserving firmness, and retaining anthocyanin content.	[80,99,103,104]
Blueberries	Chitosan	Blueberries leaf extract	Reduced microbial growth and decay rate, leading to an extended shelf-life.	[99,105]
Citrus fruit	Chitosan Gelatin	Pomegranate peel extract Cinnamaldeyde	Prolonged shelf life by inhibiting green mold development and exhibiting antifungal properties. Additionally, they reduce weight loss and decrease total acidity (TA) while enhancing total phenolic content (TPC) and antioxidant activity (AOC). Moreover, edible coatings maintain fruit firmness and glossiness throughout storage.	[80,99,100,106,107,108]
Peach	Carboxymethyl cellusose Chitosan Sodium alginate	Soybean oil Oleic acid Clorogenic acid Rhuborb extract *Aloe vera*	Regulated the degradation caused by *Penicillium expansum*, displaying antifungal characteristics, and preserved the physiological quality of the product.	[80,100,108,109,110]
Pears	Chitosan Carboxymethyl cellulose	Salicylic acid Soybean oil Oleic acid Cumin essential oils	Resulted in a reduction of PPO activity, effectively preventing the occurrence of internal browning during storage. It also minimized fungal infection, maintained fruit firmness, and significantly extended the product’s shelf life.	[99,111]
Strawberries	Sodium alginate Pectin Methyl cellulose Hydroxymethyl cellulose Chitosan Arabic gum Xanthan gum	Eugenol oil Citral essential oils *Lactobacillus plantarum* Curcumin Limonene *Asparagus* waste extract Grape seed essential oil Peony extracts Lemon essential oil	Slowed down the growth rate of molds and yeasts on the surface of strawberries. Additionally, it enhanced its functionality as a probiotic, resulting in reduced weight loss, pH, color changes, total acidity (TA), total phenolic content (TPC), and DPPH (a measure of antioxidant activity). Furthermore, the coating effectively reduced microbial growth on the strawberries.	[80,99,100,101,110,112,113,114,115]

**Table 2 foods-12-03308-t002:** Antimicrobial edible coating used in traditional sausages.

Antimicrobial Edible Coating	Active Compound	Results	References
Gelatin coating solution	_	Resulted in decreased thiobarbituric acid-reactive substances and peroxide value in traditional sausages. Additionally, it significantly reduced moisture loss by 32.6%. These findings demonstrate its effectiveness as a viable option for extending the quality and shelf life of the traditional sausages.	[120]
Chitosan	Green tea extract	The study revealed that incorporating green tea extract into the chitosan film enhanced its antioxidant and antimicrobial properties, leading to the preservation of sausage quality and prolonging its shelf life.	[121,122]
Whey protein	*Origanum virens* essential oils	The results strongly suggest that *O*. *virens* essential oil (EO) holds great potential as a food preservative for processed meat products.	[123,124]
Carboxymethyl cellulose (CMC)	Kecombrang (*Nicolaia speciosa*) extract	Coating traditional sausages with CMC effectively inhibits the growth of *Bacillus cereus*, *Escherichia coli*, *Staphylococcus aureus*, and *Pseudomonas aeruginosa*. The findings highlight CMC’s dual role as a natural antioxidant and moisture barrier coating, which contributes to preserving sausage quality and extending its shelf life.	[125]
Chitosan	Garlic and oregano essential oils	The application of the coating effectively suppressed the growth of *Salmonella* and *L. monocytogenes* bacteria while also decreasing the count of *S. aureus*.	[88]
Carboxymethyil with glycerol	Kecombrang flower extract	Resulted in a deceleration of the oxidative degradation process in traditional sausages throughout storage.	[126]
Chitosan coating with glycerol	_	Chitosan coating with glycerol was beneficial to improve the storage stability of traditional sausage at room temperature. This was evident through its ability to prevent the decline in pH, stabilize the L* value and water migration, and slow down the growth of aerobic bacteria and lactic acid bacteria.	[127]
Chitosan with sorbitol	*Medinilla spesiosa* extract	Study demonstrated that the *Medinilla speciosa* extract edible film inhibited microbiological and oxidative damages.	[128]
Whey protein isolates	Thyme, coriander, pepper, rosemary, basil	The research showcased the coatings’ ability to effectively deactivate *Listeria innocua* bacteria.	[129]
Carboxymethyil with glycerol	Kekombrang flower extract	To prevent and slow down the oxidative damage of traditional sausage products during storage.	[126]
Chitosan with sorbitol	*Portulaca oleracea* extract	Prevented both microbiological and oxidative damage in traditional sausages.	[130]
Chitosan	*_*	This study showed that among the different concentrations tested, 1% chitosan concentration proved to be the most effective in preventing bacterial growth in traditional sausages.	[131]
Cationic starch	*Origanum majorana* L. essential oil	Extended the product’s refrigerated shelf life to six days while preserving its color and odor without any impact.	[132]
Collagen with bacterial cellulose, chitosan	Essential oil	The findings of this study indicate that using the applied coating can be a viable and effective option to improve product quality and prolong its shelf life.	[133,134]

## Data Availability

Not applicable.

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
