# Peer review of "Edible Coatings and Future Trends in Active Food Packaging–Fruits’ and Traditional Sausages’ Shelf Life Increasing"

_foods, 2023, doi:10.3390/foods12173308_

Round 1
Reviewer 1 Report
Comments and Suggestions for Authors
The manuscript entitled “Edible coatings with antimicrobial activity to improve the
quality and increase the shelf life of fruits and sausages” focuses on the sustainable active packaging. Generally, the review is comprehensive, novel, and provides targeted examples. However, the following points still need to be improved:
1. Line 36, and 41, “The shelf life of a product or shelf life” should be revised as “The shelf life of a product”, and “the short shelf life or shelf life of food” should be revised as “the short shelf life of food”. Please check the manuscript and revise.
2. Line 53 and 67, “more 2 000 million more persons” should be revised as “more 2 000 million persons”, and “food chain though prevention” should be revised as “food chain through prevention”. Please check the manuscript and revise.
3. Line 90,should the format of other parts of the article be consistent when citing multiple references?Please use the same reference format.
4. Line226, please add “.” at the end of the sentence.
5. Line 364, author pointed that “Applications of edible coatings”. Title changed to Applications of antimicrobial edible coatings? Because the focus of this article is on antimicrobial edible coatings. Please check the manuscript and revise.
6. Line 384, Grammatical inconsistencies, previous research should be standardized in the past tense. Such as “decrease, increase, maintain, control, reduce” should be revised as “decreased, increased, maintained, controlled, reduced”.
7. Line 426, author pointed that “I was beneficial to improve the storage stability of smoked sausage at room temperature” should be revised as “Chitosan coating with glycerol was beneficial to improve the storage stability of smoked sausage at room temperature”. Please check the manuscript and revise.
8. Line428, this part “Recent advancements in antimicrobial food packaging” does not seem to be related to the fruits and sausages being discussed, is this somewhat inappropriate?
Comments on the Quality of English LanguageMinor editing of English language required
Reviewer 2 Report
Comments and Suggestions for Authors
The title is too long, besides I think it could be more general and not so specific
The abstract section is too long, this part needs to be improved
Keywords: please do not use the same words in the title in this section, please improve this part
Introduction
This section is confusing, in fact the authors made a too long introduction section that is really difficult to follow
The citations in all the manuscript are so different between them, it seems that they use Mendeley and they didn't check his mistakes
The sections are really confusing it´s difficult to understand the content and the principal idea of the manuscript, I suggest to analyze carefully the manuscript and making the necessary changes
Reviewer 3 Report
Comments and Suggestions for Authors
This is an interesting manuscript for improving antimicrobial activity of edible coating or film, but lots of paper discussed this topic on 2022-2024, but the author just arrange the edible coating on 2014-2021, the database was not enough. The topic and aim were not clear.
Comments on the Quality of English LanguageExtensive editing of English language required
Reviewer 4 Report
Comments and Suggestions for Authors
The subject of the article is very important, but unfortunately this review article does not bring any scientifing novelty. Many previous studies described edible, antimicrobial coatings for food aplication. The novelty of the article is expected it should be included by the authors. However, figures and tables were clearly presented. The title of the manuscript says: ‘’ Edible Coatings with Antimicrobial Activity to Improve the Quality and Increase the Shelf Life of Fruits and Sausages”. The idea is logical but after reading the review it was observed that in the case of fruits, edible coatings were described. In the case of sausages, the edible coatings were mentioned but slightly. Additionally the authors focused also on food packaging. So the section 3 of the study is not referring to the title. It has to be mentioned that the name of the genus and species of bacterial and fungal strains should be written in italics. Please double check for spelling and grammar errors. Additionally, references must be numbered in the text, placed in square brackets and listed individually at the end of the manuscript. References must be revised according to the Journal instructions for authors. My recommendation is to the major revision.
Comments on the Quality of English LanguageThe authors should double check for spelling and grammar errors
Round 2
Reviewer 2 Report
Comments and Suggestions for Authors
I only suggest adding this reference that supports the use of edible coatings on fruits https://doi.org/10.1016/j.foodcont.2022.109063
Reviewer 3 Report
Comments and Suggestions for Authors
The author had corrected the manuscript according to my comments.